# Changes in MRI Workflow of Multiple Sclerosis after Introduction of an AI-Software: A Qualitative Study

**DOI:** 10.3390/healthcare12100978

**Published:** 2024-05-09

**Authors:** Eiko Rathmann, Pia Hemkemeier, Susan Raths, Matthias Grothe, Fiona Mankertz, Norbert Hosten, Steffen Flessa

**Affiliations:** 1Institute of Radiology and Neuroradiology, University Medicine Greifswald, 17475 Greifswald, Germany or hosten@mac.com (N.H.); 2Department of Business Administration and Health Care Management, Faculty of Law and Economics, University of Greifswald, 17489 Greifswald, Germanysteffen.flessa@uni-greifswald.de (S.F.); 3Department of Neurology, University Medicine Greifswald, 17475 Greifswald, Germany; matthias.grothe@med.uni-greifswald.de

**Keywords:** machine learning integration, workflow optimization, radiological diagnostics, multiple sclerosis, report communication

## Abstract

The purpose of this study was to explore the effects of the integration of machine learning into daily radiological diagnostics, using the example of the machine learning software mdbrain^®^ (Mediaire GmbH, Germany) in the diagnostic MRI workflow of patients with multiple sclerosis at the University Medicine Greifswald. The data were assessed through expert interviews, a comparison of analysis times with and without the machine learning software, as well as a process analysis of MRI workflows. Our results indicate a reduction in the screen-reading workload, improved decision-making regarding contrast administration, an optimized workflow, reduced examination times, and facilitated report communication with colleagues and patients. Our results call for a broader and quantitative analysis.

## 1. Introduction

Multiple sclerosis (MS) is an autoimmune chronic demyelinating disorder affecting the central nervous system. The primary features of this potentially disabling disease include inflammation and neurodegeneration. MS manifests as lesions, caused by axonal or neuronal loss, demyelination, and astrocytic gliosis [1]. There are four major MS types as initially defined by the International Advisory Committee on Clinical Trials in Multiple Sclerosis in 1996 [2]: relapsing-remitting MS (RRMS), primary-progressive MS (PPMS), secondary-progressive MS (SPMS), and progressive-relapsing MS (PRMS). Patients with MS are typically diagnosed in young adulthood with a mean age of first diagnosis of 32 years [3,4]. Regular MRI measurements are of outstanding importance in the monitoring of MS patients to scan for signs of failure of the current medical regime [5], which could lead to irreversible neurologic deterioration. Despite advancements in understanding MS’ pathogenesis, there is still a lack of specific biomarkers, necessitating reliance on clinical diagnosis and imaging for patient management [6]. There are revised guidelines by three international expert groups of neurologists and radiologists to provide standardized protocols for MRI in MS diagnosis and follow-up [7]. 

MS affects about 250,000 patients in Germany [8] and 2.8 million patients worldwide (35.9 per 100,000). The pooled incidence rate of 75 reporting countries is 2.1 per 100,000 persons/year [4]. The Department of Neurology at the University Medicine Greifswald has a crucial role in the care of MS patients in North-Eastern Germany. The MS outpatient clinic cares for about 750 MS patients per year, and the neuroradiologic department carries out a large portion of the outpatients’ MR imaging. Initial and follow-up MRIs lead to a considerable amount of work hours for radiologists to guarantee adequate image acquisition, evaluation, and reporting. The augmented workload requires a closer examination of management and optimization strategies, involving working routines [9]. The rising demand and limited capacities for MR imaging require an effective workflow, which can be especially challenging in an academic environment [10]. Machine learning (ML) as a subfield of artificial intelligence (AI) is a promising promoter of simplifying working routines, especially when based on homogeneous source data [11], i.e., high-quality MRI sequences of MS patients. By implementing ML software (Version 4) into daily working routines, radiologists may be able to improve reporting accuracy, workflow efficiency, and interdisciplinary communication [12,13,14]. There are several quantitative volumetric reporting tools, which are commercially available and aim to improve radiologists’ accuracy in the interpretation of the MRI examinations of patients with multiple sclerosis [15]. Mdbrain^®^ [16,17,18,19,20,21] is a machine learning-based software which uses commonly standard MRI sequences to perform brain volume measurements, analyses of gray and white matter volumes, as well as white matter lesions. Mdbrain^®^ provides the radiologist with an automatic quantification of the lesion load of an MS patient, including a comparison with previous MRI examinations, if available. ML software support and the associated acceleration of the diagnostic process should give the radiologist more time for other tasks at hand, i.e., communication of findings, managing examination logistics, or academic/scientific challenges [22]. 

Additionally, the repeated administration of Gadolinium-containing contrast media has been generally debated [23,24]. While the use of contrast agents is essential in the initial MRI diagnosis of multiple sclerosis, their additional value in follow-up examinations is in doubt [25,26], and is considered optional, according to the current clinical guidelines in Germany. A potential benefit of administering contrast agents in follow-up MRI assessments is the identification of patients with active inflammation, as these individuals may potentially benefit from a pharmacological intervention with corticosteroids [27].

The acceptance of a new technology in a daily work routine does not only depend on medical or logistic benefits, but also on user acceptance and the transferability of new processes into existing structures [28]. To this day, the effects of the implementation of ML software on the working routines of radiologists, radiologic technicians (MR technicians), and clinicians, as well as the satisfaction of users and patients, are not fully examined [15]. This qualitative study [29] offers insights into the implementation process of the ML software mdbrain^®^ in the clinical MRI examination routine of patients with multiple sclerosis.

## 2. Materials and Methods

A qualitative investigation was conducted between November 2022 and March 2023 by the Department of Business Administration and Healthcare Management in collaboration with the Institute for Radiology and Neuroradiology at the University of Greifswald. Data collection was based on problem-centered expert interviews involving four experienced radiological residents, one neuroradiologist, one technical radiology assistant, and one neurologist, who specialized in MS treatment. In addition to questions about the process flow in routine care, the interview guideline covered aspects of professional roles, responsibilities, external factors influencing the workflow, and opinions on the use of ML software. The selection of the interview partners was based on the preliminary process analysis. The guideline for the expert interviews was divided into different sections. The first section contained questions on the interviewees’ professional group, task, and involvement in the process. In addition, the process-influencing areas of time, workload, quality, resource utilization, information, workflow, and communication were addressed. Depending on the professional group surveyed, the key questions were adapted. The interviews were documented through audio recordings and then transcribed. Through these interviews, the process pathways of MRI examinations with and without the ML software mdbrain^®^ were elucidated. Interview information was used for highlighting which parts of the daily professional routines of involved personnel would be subject to adaptation if the ML software was implemented. Key interviewee statements were deduced, summarized, and supported with quotation examples. 

After the preliminary analysis, an in-depth exploration of the MRI examination routine was undertaken through process identification and analysis. Process identification aimed to comprehensively delineate all relevant (sub-)processes associated with MRI examinations and their subsequent interpretation. The ensuing process analysis, built upon the data gleaned from process identification, sought to assess the impact of the machine learning software (mdbrain^®^) on the workflow of MRI examination and interpretation.

Mdbrain^®^ is a licensed MRI post-processing software provided by Mediaire GmbH, Berlin, Germany (https://mediaire.ai/), and certified [30,31]. Mdbrain^®^ operates as a PACS (Picture Archiving and Communication System)-integrated module. As a supporting tool for MS diagnostics, mdbrain^®^ creates two quantitative reports: a volumetry report and a lesion report. The volumetry report provides a quantitative assessment of the brain volume in comparison with a reference collective. To generate this report, mdbrain^®^ uses a three-dimensional T1-weighted gradient echo MR to segment 3 tissue classes (white matter, grey matter, cerebrospinal fluid) and 21 brain regions. Mdbrain^®^ uses a deep convolutional neural network (U-Net) trained with annotated ground truth data of more than 1000 heterogenous patient data sets from different scanner types and sequences. The volumes of these regions are then quantified, and the corresponding percentiles are calculated by comparing the measured patient’s volumes to the volumes of a healthy population (8500 healthy people) with the same covariates (age, sex, and total intracranial volume). When multiple scans of the same patient are available, a longitudinal analysis is included in the report, demonstrating possible brain atrophy dynamics over time. Volumes and percentiles are displayed in tabular format, along with clinically relevant MR slices and segmentation masks (Appendix A as example report). Additionally, a lesion characterization reporting gives a comprehensive quantitative assessment of the brain lesion load to facilitate the accurate diagnosis and subsequent monitoring of MS progression. Lesions (white matter hyperintensities) are automatically segmented from a fluid-attenuated inversion recovery sequence (i.e., FLAIR) using a deep convolutional neural network trained on the annotated data of more than 500 heterogenous patient data sets from different scanner types and sequences. After segmentation, the lesions are classified into four regions: periventricular, juxtacortical, infratentorial, and deep white matter. This classification is based on a decision tree that uses multiple image-based features as input, i.e., lesion size and location, grey/white matter ratio, form factors. The lesion count and the total lesion volume are reported for the entire brain and for each region separately. When multiple scans of the same patient are available, mdbrain^®^ performs a longitudinal analysis and classifies lesions as either “old”, “new”, or “enlarged” (Appendix B as example report). This longitudinal analysis is based on a separate deep convolutional neural network, which uses multiple image-based features of the old and the new scan [32]. Both the lesion characterization and the volumetry report are made available to the PACS within minutes after image acquisition (average processing time of volumetry within 5 min for computers with a suitable graphical processing unit). Mdbrain^®^ is an off-the-shelf, built-in software that was not specifically adapted. The training/test data of the algorithm did not contain data from the University Medicine Greifswald. Automated solutions for data transfer and the initiation of the analysis by the ML software are available (auto-pull, auto-routing). In the workflow of the University Medicine Greifswald, the radiological technician manually sent the initially acquired two MR sequences: isometric T1-w (slice thickness 1 mm, no gap, TR/TE 2100/2.5, matrix 256, FOV 250 × 250) and isometric t2_FLAIR (slice thickness 1 mm, no gap, TR/TE 7000/379, matrix 256, FOV 250 × 250) or 2D-FLAIR into the PACS, and the radiologist in charge had the choice to manually send it to a closed, local server hosting the ML software, initiating the software analysis. During the ML software analysis, further MRI sequences were acquired (DWI 4 mm, T2 sagittal 2 mm, T2 fat sat. coronal 2 mm). The radiologist then decided if additional contrast-supported sequences should be added, depending on the information of the ML report and/or conventional lesion comparison.

The interview information was contextualized with readily available information about MS MRI examinations from the hospital information system/radiology information system (HIS/RIS) software from 2019 to 2022. Parameters such as the proportion of MS MRIs with the use of the ML software, annual examination frequencies, MRI machines/field strength, homogeneity of utilized MRI sequences, and mean duration of measurement protocols were systematically obtained from HIS (Version 7.2.2) [33]/RIS (Version 70.0.15115.0) [34] software. There were documented time stamps for the MRI sequences and the final proofreading of the diagnostic report, but not for the duration of the initial creation process for the diagnostic report. Since there were no readily available data about the written report times, we conducted an exemplary time measurement of the diagnostic evaluations by medical personnel without the assistance of a machine learning software, which involved assessments by 5 radiologists (1 neuroradiologist and 4 experienced radiological residents with >5 years of professional experience). From a pool of 25 randomly chosen complex MS MRIs with at least 10 lesions, each physician was assigned to evaluate the T1- and FLAIR-images of 5 randomly chosen MRIs and their respective preliminary examinations. Due to randomization, 4 MRIs were analyzed twice (by different radiologists) and 4 were not chosen at all. Accordingly, a total of 21 assessment times were recorded by 2 less experienced radiologists (<1 year of professional experience) without ML support. The documented times of the radiologists to compare the T1 and FLAIR sequences of the current MRI with those of the respective preliminary examination were measured. The goal for the radiologists was to capture all the relevant pathologies necessary for a written report and, if necessary, to recommend supplementary contrast administration. For comparison, we measured the times it took for two inexperienced radiologists (<1 year of professional experience) to capture all the relevant information from each of the conventionally analyzed MRIs with the assistance of ML software. Since the same MRIs were assessed with and without ML support, paired t-tests with SPSS Statistics (version 29.0.0.0; IBM, Armonk, NY, USA) were performed to test for significant differences in the mean assessment times (significance level = 0.05). The participating personnel were familiar with the conventional as well as with the ML software-supported way of evaluating MS MRIs. In the daily work routine, the radiologists are advised to control all ML findings for potential errors. Additionally, all findings (including those implemented through ML) are analyzed and corrected if necessary by a neuroradiologist before final approval. We used contrast agent administration rates, based on billing data, for a comparison of the proportion of examinations, in which contrast agents were administered.

## 3. Results

To answer the question of how the introduction of the ML software has influenced the workflows of radiologists, radiologic technologists (MR technicians), and clinicians, we initially aimed to objectify the workflows through process visualization and descriptive analyses. Between 2019 and 2023, an average of 352 MRI head examinations of MS patients per year were performed, for MS purposes alone. Data homogeneity was good with >80% of the examinations performed at 3T field strength (MAGNETOM Vida and MAGNETOM Skyra Fit), and a standardized sequence protocol for MS examinations, only deviating in initial examinations, with additional optic nerve sequences. The ML software mdbrain^®^ was gradually implemented in the workflow, starting in the last quarter of 2021. The usage of the product by the attending personnel was optional. The portion of MS examination with a report from the ML software rose within 3 months to over 85 percent. 

Figure 1 and Figure 2 show the respective MRI workflow with and without the use of the ML software. Decision situations with and without contrast agent administration are shown in each case. If the patient has the first MS MRI or describes a change in symptoms since the last follow-up MRI, an intravenous line is inserted right after the patient has been informed and given consent. First, native images (T1 and FLAIR) and contrast-enhanced MRI images are acquired sequentially. At the end of the MRI examination, the patient is discharged. Depending on the radiologist’s availability, the MRI images are analyzed, comparing the newest images with previous findings. The MRI examination is finalized by writing the report. An interim evaluation of the MRI images does not usually take place. If the patient does not initially describe any change in symptoms for an MRI examination without the ML tool (Figure 1), only native MRI images are initially taken after the patient gives informed consent. After the images (T1 and FLAIR) have been transferred to the PACS system manually by the MR technician, the radiologist analyzes the images compared to previous lesion sizes and numbers, and possible signs of new or progressive atrophy. During this time, the patient remains in the MRI scanner, where further sequences are acquired. If there are any abnormalities, further contrast-enhanced images are supplemented. The crucial time factor at that point is that a radiologist must first be available to carry out a comprehensive assessment of the images. If the radiologist is busy with other examinations or other clinical activities, there will be waiting times for the MRI patients and staff. 

For the workflow using the ML software (Figure 2), a check is carried out before the patient enters the MRI examination to determine whether previous findings from earlier examinations are already available. If this is the case, the radiology assistants check whether these findings have already been analyzed using the ML tool. If not, this is conducted subsequently. The corresponding report is then already available for comparison for the upcoming examination. Once the patient has given their informed consent, native MRI images are taken, as in the classic workflow. The radiological technician manually sends the initially acquired MRI sequences (isometric T1-w, and isometric t2_FLAIR or 2D-FLAIR) to the PACS, and the radiologist in charge manually sends it to the closed, local server hosting the ML software, initiating the software analysis. A data transfer to external servers is not necessary. There are already automation solutions for this process step that prevent potential waiting times due to the pending transfer. Automation solutions for this process step have not yet been implemented at the Institute for Radiology and Neuroradiology at the University of Greifswald. The MRI images are then analyzed, and the lesion report is generated by the ML tool. This usually takes less than five minutes and therefore covers approximately the time the patient remains in the MRI scanner for further images anyway. The lesion report is used as the basis for a radiological assessment and a decision on the need for further contrast-enhanced images. Using the ML tool seems to offer especially a reduction in the time to decide on further contrast-enhanced images. The radiologist must now review the lesion report, in which potential changes to previous examinations are already identified by the ML tool. The time required to evaluate the native MRI images and compare them with previous findings could be reduced compared to the classic approach, depending on the time intensity of conventional comparisons.

Following the introduction of the ML software, radiologists, MR technicians, and physicians emphasized the following five aspects of its use in routine care:Workload and Efficiency:

The integration of the ML software into the MS MRI workflow simplified the decision-making process. By directing attention to specific lesions/areas of interest, the ML report simplified the manual examination process. The routine task of lesion counting was shifted to the ML software, which freed up radiologists to focus on other tasks at hand. It was mentioned that the algorithm also showed promise in being more sensitive in detecting lesions, especially in complex cases with a large lesion count. It was suggested that the ML software might speed up decision-making, helping physicians to decide on further actions, like administering contrast agents or changing the therapeutic regime.

**Citation 1.** *“The process of decision-making is made significantly [...] less complicated. [The machine learning software] directs you as to which lesions to examine. [...] the task was shifted from ‘compare each and every spot’ to ‘check whether [the machine learning software] is correct in its assessment’.” (neuroradiologist)*

**Citation 2.** *“[...] For example, those lesions’ volumes are measured by [the machine learning software], which is simply impossible utilizing only examining them manually. [...] it, of course, alleviates having to do routine counting [of the lesions]. Whether they are really 52 or 54 small [lesions] in the brain, which you’d have to arduously count otherwise. […] It’s simply manual work which is taken over [by the machine learning software], and one therefore has more capability for other areas of work, which may require more physician expertise than counting lesions one by one.” (radiologist 1)*

2.Systematic Errors vs. Human Interpretation:

The errors made by the ML software were perceived as rather systematic, potentially differing from the variations in an individual radiologist’s interpretation or the inter-observer differences of different radiologists. The algorithm apparently showed a high negative predictive value, making it potentially beneficial for detecting lesions and accurately identifying the absence of new ones.

**Citation 3.** *“From my perspective, one would methodologically describe it as having a high negative predictive value. This algorithm detects many lesions that may not actually exist, but when the algorithm indicates ‘there is no new lesion’, there typically is indeed no new lesion. […] I always find that, at a qualitative level, there is a significant distinction when it comes to errors; [errors made by the machine learning software] tend to be systematic in nature. This is in contrast to a radiologist who might have good and bad days. The quality of their interpretation may differ when the radiologist writes the report first thing in the morning versus [...] at 10:30 p.m. in the evening. When [a patient] has 173 lesions, and now you’re [required] to find the 174th, it’s not ideal. In such cases, an algorithm ultimately proves to be not only faster but also more sensitive.” (neurologist)*

3.Lesion Analysis and Communication:

The ML software could make quantitative lesion analysis easier, providing information that might be challenging or impossible to obtain manually. The software might enable more precise location descriptions of new lesions, which could improve communication between doctors and patients. Especially post-examination communication was reportedly faster and more efficient. The reports generated by the ML software were reported to be visually appealing and easily accessible to patients, potentially enhancing the quality of communication in doctor–patient interactions. This applies not only to the initial diagnosis of MS, but also to the context of the course of the disease. The reports can be used to transparently communicate the MRI recognizable course of the disease and make any new clinical symptoms that may have emerged easier to understand. In a competitive environment, an improved patient loyalty was considered plausible.

**Citation 4.** *“So, [the lesion load] is presented in a way that’s easy for patients to understand, which I think is excellent. I therefore believe the quality of communication between doctor and patient is improved through it. […] Communication between doctor and patient, in my opinion, seems to have decreased. This is because [communication] is not necessary in every case now. But communication now is perhaps a bit more targeted. […] I find the presentation of this report, as it is visually designed, to be very appealing, and it is some-thing that is highly accessible to patients.” (neurologist)*

**Citation 5.** *“Well, there’s less communication between doctor and patient before the MRI examination. However, if desired, communication after the examination is much faster and easier. […] The volume of the lesions measured by the machine-learning software is something I hadn’t previously noticed. [...] Measuring the volume of each individual lesion manually would be unrealistic and not feasible during the course of a shift.” (radiologist 2)*

4.Time Savings and Workload Distribution:

The utilization of machine-based software products for lesion assessment enables radiologists to make quick and flexible decisions regarding the necessity of contrast-enhanced imaging studies. In our sample analysis (Table 1), the experienced radiologists (>5 years of training) required an average of 296 s per MRI to capture all the relevant contents for complex MS MRIs (w/o ML). Thereby, the time spent is independent of the number of lesions; there is no statistical correlation (Figure 3). With the support of the ML software, the inexperienced radiologists (<1 year of training) took an average of 82.4 s to capture all the relevant contents in the same MS MRIs (avg. w ML). When using ML, the assessment time to capture all the relevant information from an MS MRI is significantly shorter; on average, the time for the assessment is reduced by 210 s. In addition, as can be seen from the boxplot, the spread of the examination times is significantly narrower when ML is used.

Regardless of the measurement times, the perception of time savings was confirmed by the process participants interviewed:

**Citation 6.** *“Additionally, the time it takes for the program to generate a report for me is roughly five minutes. [...] It would take [a reporting radiologist] significantly longer to manually do it at this level of thoroughness and conscientiousness. […] While the program is running in the background, the reporting radiologist is able to work, and then make an ad-hoc decision whether further MR sequences are necessary. This definitely saves a significant amount of time.” (radiologist 2)*

**Citation 7.** *“When new lesions are assessed by the machine-learning software, you can provide a precise description of their location, which wouldn’t be feasible with manual assessment. […] If you are familiar with the colors [which highlight each lesion’s location], the reporting radiologist can assess the report created by the program and form a preliminary result within less than ten seconds. Because it’s color-coded, you can immediately tell if a lesion is new or known. In clinical routine, the longest part [of the process] is usually waiting for the program’s report.” (radiologist 1)*

5.Contrast Media

We did not find a significant reduction in the number of contrast-enhanced examinations after the implementation of the ML software, comparing the fraction of contrast-supported MRI examinations from 2019 to 2023. However, it must be emphasized that a new MRI guideline for MS was published during the period under review, which now recommends the avoidance of contrast agents in the follow-up examinations of MS patients. Against this background, the effects of the now no longer recommended use of contrast media overlap with the faster and more objective decision for or against the use of contrast media based on the ML report.

**Citation 8.** *“[...] By allowing us to evaluate lesion load simultaneously [to the MRI], AI enables us to be more flexible in decision-making on whether we want to add contrast-enhanced sequences. [This] is advantageous in avoiding unnecessary placement of intravenous cannula and administration of medication.” (radiologist 1)*

**Citation 9.** *“The program quickly helps me decide whether to administer the contrast agent or not, and I believe this really does reduce the use of contrast-agent in general.” (physicians 2)*

6.Drawbacks and Difficulties

The procurement and installation of a server specifically configured for this purpose were necessary. Furthermore, coordination with the legal department, IT department, and data protection was required.

The interviewees mentioned the weaknesses of the ML software in the posterior cranial fossa and generally in artifact-prone areas. 

**Citation 10.** *“The ML software usually detects a little more than I do, but sometimes those are lesions that aren’t actually real; they’re just MRI artifacts that occur. So, I critically examine them to determine whether they’re genuine or not.” (radiologist 2)*

**Citation 11.** *“We had initial issues with the lesions located in the posterior cranial fossa. It’s already a bit better. It’s not yet optimal, but it has improved. So, I know I need to check ML findings here.” (radiologist 1)*

## 4. Discussion

The implementation of the ML software in the MRI examination workflow for patients with multiple sclerosis (MS) seems to yield several advantageous effects. Introduced as an optional support software for the radiologists in charge, the ML software was accepted rapidly, becoming the preferred way of analyzing an MS MRI. The interviewed radiologists highlighted a reduction in the screen-reading workload as a reason, perceiving manual lesion counting as a laborious task. The interviewees described a shift from time-consuming lesion comparisons to rather checking predefined results, allowing a reallocation of resources towards other tasks at hand. The reduction in the workload was accordingly pronounced in cases involving complex findings with a high number of lesions or complexly confluent lesions, which are more challenging to compare with the classic approach of lesion counting [35]. The adoption of the ML software seems to create significant time savings for the radiologist, reportedly taking approximately five minutes compared to a considerably longer time for manual reporting. A reduction in screen reading times seems to be confirmed in our sample of report times, in which inexperienced radiologists with ML software were 3–4 times quicker than experienced radiologists without, and comparably quick to a neuroradiologist without ML software. Furthermore, the range of times required when using ML is significantly narrowed, which considerably facilitates the planning of personnel deployment in the long term. The background operation of the software allowed radiologists to work simultaneously. The accuracy of the ML software was described as reliable. One radiologist even stated that human radiologists could not possibly achieve the level of precision of the ML software in complex lesion patterns. However, it is essential to further objectify the reliability of using the ML software in this context, as missed lesions could delay necessary alterations in therapeutic regimes aimed at preventing neurologic symptoms and functional deterioration. An improvement in the precision and comparison of the lesions in radiological reports seems likely, and thus supports the results provided in the study by Barnett et al. on the comparability of ML-based evaluations [36]. In the radiologists’ experience, the usage of the ML software tended to lead to a higher sensitivity and a lower specificity, due to often artifact-related false-positive findings. The interviewees described the typical errors of the ML software as false positive and rather systematic, in a way, that they often occurred in artifact-burdened regions of the brain, i.e., close to the skull base. Those false findings might be easier to identify since they are typically located. In comparison, the variations in a radiologist’s interpretation might be more diverse, being influenced by factors like workload and fatigue towards the end of a workday. By optimizing workflow and curtailing examination durations, the software could enhance efficiency, allowing for the examination of a greater number of patients within a given timeframe. This improved cost-efficiency is appealing to healthcare facilities aiming to increase patient throughput. Moreover, it could shorten patients’ waiting times in situations of relative MRI scarcity. The influence of the ML software’s role on decision-making processes regarding the necessity of contrast application might be noteworthy as well. Although there is no evidence of contrast media residues with macrocyclic contrast media, the current MS guidelines call for a well-reasoned approach if contrast media are necessary or not. Optimizing contrast applications could contribute to material and time conservation and the objective of mitigating potential patient risks associated with the administration of contrast media. In the future, the implementation of additional sequences might additional value to ML software performance. The central vein and peripheral rim signs are examples of promising findings in susceptibility-weighted imaging (SWI), which could further improve the accuracy of ML software assessment [37,38,39,40]. The generation of structured and easily comprehensible reports by the ML software facilitates communication between radiologists, patients, and clinicians. The lesion report generated by the ML software was highlighted as beneficial by the neurologist, as it provided a comprehensive overview of the development of brain lesions. Changes in the lesion load or atrophy could thus be easily assessed by non-radiological personnel and patients, leaving only borderline findings open for radiologic feedback. The reports facilitated the communication of findings, giving the patients understandable information about their current disease burden, and possibly necessary alterations of the therapeutic regime. The report may also foster physician–patient relationships and improve patient loyalty, especially in an out-patient setup.

Limitations: The results reflect the experiences of a limited number of employees from a single MS-treating facility, and therefore do not allow for universally applicable conclusions. In particular, the representation of the patient’s perspective is based solely on the secondary depiction provided by the treating neurologist, although studies underline the patient’s high interest in MRI education [41,42]. The validity of ML software lesion counts should be further objectified and compared to human precision. Although human lesion analysis remains the gold standard, superior human performances in highly complex MS examinations seem doubtful and other study results already indicate this [36,42,43]. While the study aimed to assess the general impact of the mdbrain^®^ software, the assessment was primarily focused on depictions of the diagnostic and workflow benefits by medical personnel. The report time measurements involved a small sample size, which may limit the generalizability of the findings. However, this result is consistent with the results of a study on ML-assisted breast cancer diagnostics, which showed a significantly lower workload with a similar cancer detection rate [44]. Other aspects, such as communication patterns, decision-making processes, and overall efficiency, may not have been fully captured. The contextualization of interview information with historical data from the HIS/RIS software might not fully reflect current practices or future trends in MS MR examinations. Mdbrain^®^ is a proprietary software and can thus not be fully described methodically. The use of commercial software for transcription may introduce errors or biases during the transcription process. Additionally, the accuracy and completeness of the historical data could impact the validity of the comparisons made. For example, the approach to use billing data to estimate changes in the use of contrast media was abandoned due to data inconsistencies. The report time measurements involved a small sample size, which may limit the generalizability of the findings.

## 5. Conclusions

In conclusion, our study suggests that the integration of ML software products such as mdbrain^®^ in an MRI workflow offers opportunities to streamline processes, enhance efficiency and precision, and improve communication with other physicians and patients. However, it is essential to acknowledge the limitations and uncertainties associated with our findings and continue exploring the broader implications of such implementations in further studies.

## Figures and Tables

**Figure 1 healthcare-12-00978-f001:**
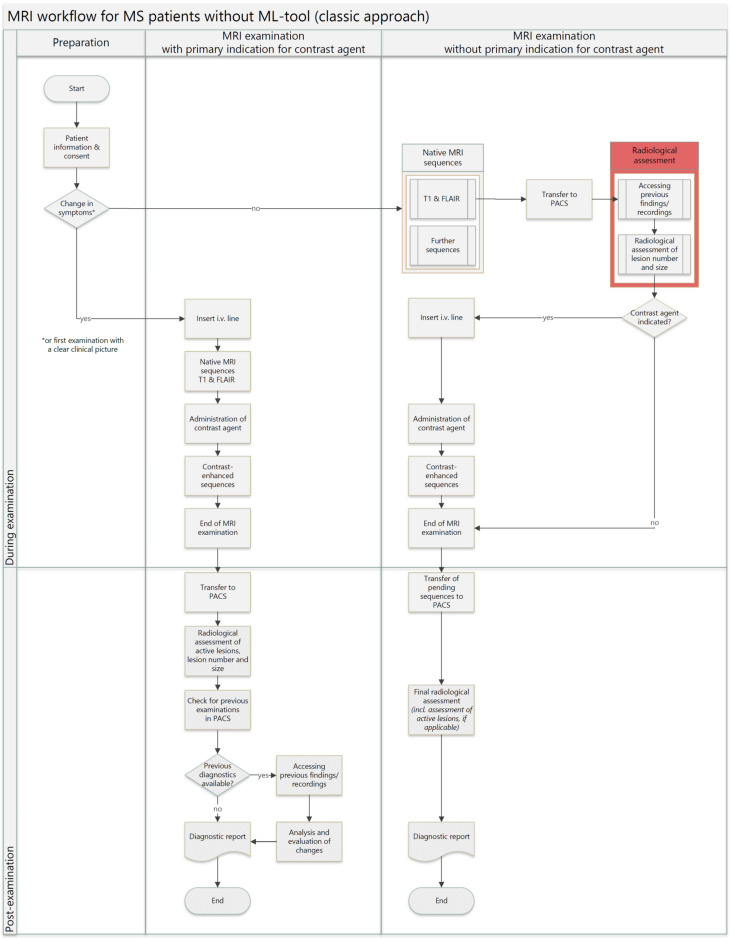
MS MRI workflow without ML tool.

**Figure 2 healthcare-12-00978-f002:**
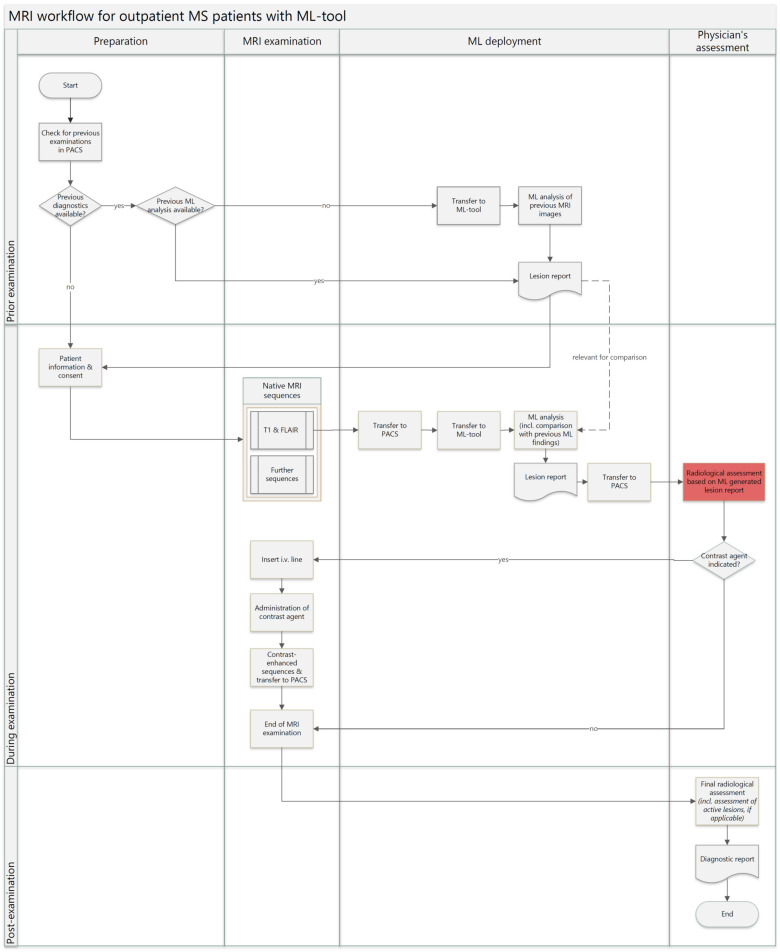
MS MRI workflow with ML tool.

**Figure 3 healthcare-12-00978-f003:**
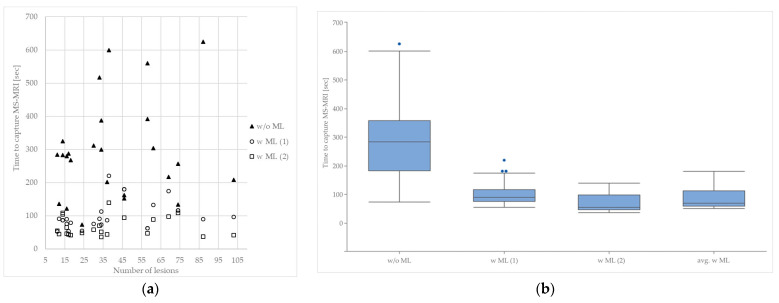
Influence of the utilization of machine learning on the time needed to capture relevant data per MS MRI, n = 25. (**a**) Time to capture MS MRI depending on the number of lesions. (**b**) Time to capture an MS MRI with and without the use of ML (w/o ML: without machine learning; w ML (1): with utilization of machine learning, radiologist 1; w ML (2): with utilization of machine learning, radiologist 2; avg. w ML: average time to capture an MS MRI with utilization of machine learning).

**Table 1 healthcare-12-00978-t001:** Time for the review of MS MRIs with and without ML.

Variable	n	Mean	SD	Min	Median	Max	∆Mean (95% CI)	*p*-Value
Number of lesions	21	39.62	26.39	11	34	103	-	-
w/o ML [s]	25	295.5	149.2	73.0	283.0	625.0	0	-
w ML (1) [s]	21	101.60	44.32	53.00	90.00	221.00	193.88 ** (130.10–257.66)	<0.001
w ML (2) [s]	21	68.80	29.94	36.00	55.00	139.00	226.68 ** (163.88–289.48)	<0.001
avg. w ML [s]	21	82.40	34.70	49.00	68.00	180.00	210.28 ** (147.30–273.26)	<0.001

** Significant mean difference compared to w/o ML. (w/o ML: without machine learning; w ML (1): with utilization of machine learning, radiologist 1; w ML (2): with utilization of machine learning, radiologist 2; avg. w ML: average time to capture an MS MRI with utilization of machine learning).

## Data Availability

The raw data supporting the conclusions of this article will be made available by the authors upon request.

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
