# Peer review of "Changes in MRI Workflow of Multiple Sclerosis after Introduction of an AI-Software: A Qualitative Study"

_healthcare, 2024, doi:10.3390/healthcare12100978_

Round 1

Reviewer 1 Report

Comments and Suggestions for Authors

The authors demonstrated that using an AI software can help daily radiological diagnostics. The authors quantified their arguments by comparing the processing times with and without machine learning assistance. I have the following comments that may help the authors, but if any of them are not relevant or wrong, feel free to ignore them:

1.       For the Materials and Methods section, the authors described the process of using the software Mdbrain to process MRI images with/without ML methods. However, the authors never really talked about what kind of ML algorithms and methods they used. I understand that it is probably some built-in algorithms within the software. However, it is still valuable to have some information about the options of the ML algorithms the software offers and what specific algorithms the authors used. Are there fine-tuned options for the chosen algorithm?

2.       Again, about the ML algorithm: do the users/authors need to train an ML model, or can we directly use the built-in model? If the authors need to train the model, what kind of data and how the model was trained should be described. If the authors used the built-in model, the authors should also talk about how the software company developed the model and whether the model is generalized and powerful enough to handle all kinds of specific image segmentation/recognition needs.

3.       Line 196 and the text in the results section: the authors claimed that the software can handle images in less than five minutes. Can the authors mention the usual image size in terms of pixel size? Large images normally take a longer time to process.

4.       Line 301: Table 1 and Figure 3 show the same information. I recommend keeping only one of them.

5.       The authors mainly compared the processing time between methods with and without machine learning. I recommend the authors also describe/quantify how accurate is the machine learning method compared to manual inspection.

6.       I recommend adding a very short description of your supplementary materials section instead of only showing the Lesion report.

Author Response

Dear reviewer 1, Dear ‘healthcare’ team,

we hope this letter finds you well. We are writing to submit the revised version of our manuscript “Changes in MRI Workflow of Multiple Sclerosis after Introduction of an AI-software: a qualitative study", which was previously submitted for consideration to healthcare. We appreciate the opportunity to address the feedback provided by the reviewers. My team and I would like to express our gratitude for the constructive comments and suggestions offered by the reviewers. We have carefully reviewed the remarks and suggestions. As there were overlaps in the points of critique, we have consolidated the addressed issues into a single shared document. We hope this meets with your approval. In the attached PDF file is a modified manuscript version with marked text changes. In response to the feedback, we have meticulously revised the manuscript, incorporating the following changes:

  1. Mdbrain (machine learning-algorithm, development/training); reviewers 1, 2, 4, 5:

We have contacted Mediaire GmbH and requested more detailed information about the development and testing of the ML (machine learning) software mdbrain. The methods section has been supplemented with this information (line 104-135), and a corresponding acknowledgment of this contribution has been inserted into the acknowledgments section. We kindly request to consider that mdbrain is a commercial software, and due to proprietary reasons, deeper insights into its development are currently not available.

  1. Mdbrain options/fine-tuning; reviewer 1:

There were no further configuration options for the user. We added comment for clarity in line 138-139.

  1. Staff training; reviewer 4:

There was no dedicated training of the staff for this study. The staff was however familiar with the analyzing process with and without ML software in the clinical working routine (line 180-182).

  1. Pixel size (line 115-117); reviewer 1:

Pixel sizes can be calculated with the matrix size and FOV provided. If we should provide it, despite its redundancy, please let us know.

  1. Time of ML analysis <5 min; reviewer 1:

We can confirm this time frame for volumetry (specified in manuscript). According to the manufacturer's specifications, the evaluation time varies depending on the technical setup, such as server specifications, especially the built-in graphics processor, and network capacities. The pure export process from the PACS to the local server takes approximately 1:25 minutes. Subsequently, there is a configurable latency period during which the server waits for any subsequent input (currently set to 30 seconds), so that the analysis process starts approximately 2:05 minutes after the initial send command. On the universally available web interface, one obtains the volumetric result after about 4:15 minutes, the lesion report including longitudinal comparison after approximately 7:18 minutes. The mirroring of the report sheets into the PACS is completed after approximately 8:00 minutes.

  1. Table 1 and Figure 3 showing the same information (line 301); reviewer 1:

With the plot of time versus number of lesions, we want to make clear that there is no correlation between the number of lesions and reporting time, neither when using ML, nor without using ML. It also demonstrates, that a high lesion count leads to a longer assessment time. As you point out, we show data that is redundant in the table and box plot. We w consider the box plot to be a very impressive image to emphasize the significant difference in time between diagnosis with and without the use of ML. We believe that it enables the readers to grasp the core message at a glance.

  1. Accuracy of ML vs. ‘gold standard’ or benchmark; reviewers 1, 2, 4:

We share the appreciation for a quality comparison between human-based findings and those generated by ML, and we aim to conduct such a quality assessment for a subsequent publication.

  1. Lack of validation of ML-findings; reviewer 2:

Please take into consideration that this is a qualitative study with a different approach. In our daily work routine, all findings (including those implemented through ML) are analyzed by a neuroradiologist before final approval and corrected if necessary. The objective validation of these results is still pending and will be published at the end of this year.

  1. Statistical tools rationale; reviewer 4:

Since we analyzed paired measurement data, the sample size was larger than 20 and the data in the Q-Q diagram showed a good approximation to a normal distribution, we opted for a parametric test - the paired t-test - to analyze the mean differences between the measurement times with and without the use of ML. In order to assume a deviation from the normal distribution requirement of the paired t-test, we also carried out non-parametric tests for paired samples (Wilcoxon signed-rank test and sign test), that also lead to a significant difference in the median value between the findings with and without ML. We have added a passage on the selection of the paired t-test, and thus hope to have explained the test selection more transparently (line 174-176).

  1. Abbreviations; reviewer 3:

We have added an abbreviation list and included it in the supplementary materials. However, we are open to positioning it elsewhere if preferred.

  1. Time presentation; reviewer 3:

We have now standardized the information in minutes to seconds.

  1. Commas in table 1; reviewer 3:

We apologize for the confusion. In German, the comma marks the decimal places and as our statistic programs work with German input values, outputs are also produced with commas. We did a reformatting into the English conventions and have now adjusted the numerical data in the text and table accordingly.

  1. Different numbers in Table 1 for the ML and non-ML group; reviewer 3:

In the non-ML group 5 radiologists analyzed 5 MRIs each randomized out of a pool of 25 complex MRIs. To prevent selection-bias, we randomized the MRIs for each radiologist. Due to randomization, 4 MRIs were analyzed twice (by different radiologists) and 4 were not chosen at all. The ML group analyzed the MRIs from the non-ML group, so they had only 21 MRIs to analyze. We have now added a passage to the methodology to better clarify the difference in the number of time measurements between MRIs with and without ML (line 164-169).

  1. Resolution of the figure (Figure 3); reviewer 3:

We have seen that the resolution of the images in the .pdf is indeed very poor. We have inserted the images again in higher resolution in the Word document and checked the resolution, when transferring them to a pdf-file. We hope that the images are now actually displayed in a better quality. Please let us know, if there are remaining image quality issues.

  1. Interview selection criteria; reviewer 4:

Paragraph provided (line 92-97).

  1. MDPI format issues; reviewer 5:

The manuscript was created according to the ‘healthcare’-template provided. The categories from the template were utilized.

  1. Display of dataset/data collection; reviewer 5:

The primary data collection consisted of the described interviews (qualitative study). We do not consider a graphical representation to be useful in this context.

  1. Drawbacks or difficulties in using the ML data flow; reviewer 4:

There weren't many difficulties, apart from the initial server setup and the associated administrative challenges. Potential vulnerabilities of the ML software must be considered in the final assessment, as mentioned already in the manuscript. Additional small paragraph provided (line 375-387).

  1. Some figures being put in paper and some in supplementary materials; reviewer 5:

We cannot comprehend the confusion. The main graphics are in the main manuscript, and supplementary graphics, such as the sample reports, are in the supplements.

  1. Enhanced clarity/readability; reviewer 2:

In addition to the aforementioned points, we have attempted to enhance clarity and readability. If there are still unclear passages or ambiguous phrasings in your opinion, we kindly ask you to specifically point them out, and we will be happy to make improvements.

We hope that the revised version aligns with the standards of excellence upheld by healthcare and will be of interest to your readership.

Thank you once again for the opportunity to revise and resubmit our manuscript. I look forward to hearing from you regarding the outcome of the review process. Please do not hesitate to contact me if you require any further information or clarification.

Sincerely,

Eiko Rathmann

Reviewer 2 Report

Comments and Suggestions for Authors

The study, while presenting valuable insights into the integration of machine learning (ML) software into MRI workflows for MS patients, lacks comprehensive validation of the reliability and accuracy of the ML software's lesion analysis. 

Without robust comparative analysis against established gold standards or existing benchmark tests, the study's conclusions regarding the efficacy and precision of the ML software may be unsubstantiated. 

However, study lacks thorough exploration and discussion of technical aspects related to the implementation of the ML software, such as the specifics of the algorithm used, potential biases in the training data, or the transparency of the decision-making process within the software. Without this critical technical insight, readers are left with an incomplete understanding of the software's functioning and limitations, hindering their ability to assess the validity and generalizability of the findings.

This omission undermines the reliability of the findings and diminishes the scientific rigor necessary for publication.

Comments on the Quality of English Language

The quality of English language in the manuscript is generally good. However, there are several areas where improvement is needed to enhance clarity and readability.

Author Response

Dear reviewer 2, Dear ‘healthcare’ team,

we hope this letter finds you well. We are writing to submit the revised version of our manuscript “Changes in MRI Workflow of Multiple Sclerosis after Introduction of an AI-software: a qualitative study", which was previously submitted for consideration to healthcare. We appreciate the opportunity to address the feedback provided by the reviewers. My team and I would like to express our gratitude for the constructive comments and suggestions offered by the reviewers. We have carefully reviewed the remarks and suggestions. As there were overlaps in the points of critique, we have consolidated the addressed issues into a single shared document. We hope this meets with your approval. In the attached PDF file is a modified manuscript version with marked text changes. In response to the feedback, we have meticulously revised the manuscript, incorporating the following changes:

  1. Mdbrain (machine learning-algorithm, development/training); reviewers 1, 2, 4, 5:

We have contacted Mediaire GmbH and requested more detailed information about the development and testing of the ML (machine learning) software mdbrain. The methods section has been supplemented with this information (line 104-135), and a corresponding acknowledgment of this contribution has been inserted into the acknowledgments section. We kindly request to consider that mdbrain is a commercial software, and due to proprietary reasons, deeper insights into its development are currently not available.

  1. Mdbrain options/fine-tuning; reviewer 1:

There were no further configuration options for the user. We added comment for clarity in line 138-139.

  1. Staff training; reviewer 4:

There was no dedicated training of the staff for this study. The staff was however familiar with the analyzing process with and without ML software in the clinical working routine (line 180-182).

  1. Pixel size (line 115-117); reviewer 1:

Pixel sizes can be calculated with the matrix size and FOV provided. If we should provide it, despite its redundancy, please let us know.

  1. Time of ML analysis <5 min; reviewer 1:

We can confirm this time frame for volumetry (specified in manuscript). According to the manufacturer's specifications, the evaluation time varies depending on the technical setup, such as server specifications, especially the built-in graphics processor, and network capacities. The pure export process from the PACS to the local server takes approximately 1:25 minutes. Subsequently, there is a configurable latency period during which the server waits for any subsequent input (currently set to 30 seconds), so that the analysis process starts approximately 2:05 minutes after the initial send command. On the universally available web interface, one obtains the volumetric result after about 4:15 minutes, the lesion report including longitudinal comparison after approximately 7:18 minutes. The mirroring of the report sheets into the PACS is completed after approximately 8:00 minutes.

  1. Table 1 and Figure 3 showing the same information (line 301); reviewer 1:

With the plot of time versus number of lesions, we want to make clear that there is no correlation between the number of lesions and reporting time, neither when using ML, nor without using ML. It also demonstrates, that a high lesion count leads to a longer assessment time. As you point out, we show data that is redundant in the table and box plot. We w consider the box plot to be a very impressive image to emphasize the significant difference in time between diagnosis with and without the use of ML. We believe that it enables the readers to grasp the core message at a glance.

  1. Accuracy of ML vs. ‘gold standard’ or benchmark; reviewers 1, 2, 4:

We share the appreciation for a quality comparison between human-based findings and those generated by ML, and we aim to conduct such a quality assessment for a subsequent publication.

  1. Lack of validation of ML-findings; reviewer 2:

Please take into consideration that this is a qualitative study with a different approach. In our daily work routine, all findings (including those implemented through ML) are analyzed by a neuroradiologist before final approval and corrected if necessary. The objective validation of these results is still pending and will be published at the end of this year.

  1. Statistical tools rationale; reviewer 4:

Since we analyzed paired measurement data, the sample size was larger than 20 and the data in the Q-Q diagram showed a good approximation to a normal distribution, we opted for a parametric test - the paired t-test - to analyze the mean differences between the measurement times with and without the use of ML. In order to assume a deviation from the normal distribution requirement of the paired t-test, we also carried out non-parametric tests for paired samples (Wilcoxon signed-rank test and sign test), that also lead to a significant difference in the median value between the findings with and without ML. We have added a passage on the selection of the paired t-test, and thus hope to have explained the test selection more transparently (line 174-176).

  1. Abbreviations; reviewer 3:

We have added an abbreviation list and included it in the supplementary materials. However, we are open to positioning it elsewhere if preferred.

  1. Time presentation; reviewer 3:

We have now standardized the information in minutes to seconds.

  1. Commas in table 1; reviewer 3:

We apologize for the confusion. In German, the comma marks the decimal places and as our statistic programs work with German input values, outputs are also produced with commas. We did a reformatting into the English conventions and have now adjusted the numerical data in the text and table accordingly.

  1. Different numbers in Table 1 for the ML and non-ML group; reviewer 3:

In the non-ML group 5 radiologists analyzed 5 MRIs each randomized out of a pool of 25 complex MRIs. To prevent selection-bias, we randomized the MRIs for each radiologist. Due to randomization, 4 MRIs were analyzed twice (by different radiologists) and 4 were not chosen at all. The ML group analyzed the MRIs from the non-ML group, so they had only 21 MRIs to analyze. We have now added a passage to the methodology to better clarify the difference in the number of time measurements between MRIs with and without ML (line 164-169).

  1. Resolution of the figure (Figure 3); reviewer 3:

We have seen that the resolution of the images in the .pdf is indeed very poor. We have inserted the images again in higher resolution in the Word document and checked the resolution, when transferring them to a pdf-file. We hope that the images are now actually displayed in a better quality. Please let us know, if there are remaining image quality issues.

  1. Interview selection criteria; reviewer 4:

Paragraph provided (line 92-97).

  1. MDPI format issues; reviewer 5:

The manuscript was created according to the ‘healthcare’-template provided. The categories from the template were utilized.

  1. Display of dataset/data collection; reviewer 5:

The primary data collection consisted of the described interviews (qualitative study). We do not consider a graphical representation to be useful in this context.

  1. Drawbacks or difficulties in using the ML data flow; reviewer 4:

There weren't many difficulties, apart from the initial server setup and the associated administrative challenges. Potential vulnerabilities of the ML software must be considered in the final assessment, as mentioned already in the manuscript. Additional small paragraph provided (line 375-387).

  1. Some figures being put in paper and some in supplementary materials; reviewer 5:

We cannot comprehend the confusion. The main graphics are in the main manuscript, and supplementary graphics, such as the sample reports, are in the supplements.

  1. Enhanced clarity/readability; reviewer 2:

In addition to the aforementioned points, we have attempted to enhance clarity and readability. If there are still unclear passages or ambiguous phrasings in your opinion, we kindly ask you to specifically point them out, and we will be happy to make improvements.

We hope that the revised version aligns with the standards of excellence upheld by healthcare and will be of interest to your readership.

Thank you once again for the opportunity to revise and resubmit our manuscript. I look forward to hearing from you regarding the outcome of the review process. Please do not hesitate to contact me if you require any further information or clarification.

Sincerely,

Eiko Rathmann

Reviewer 3 Report

Comments and Suggestions for Authors

Rathmann et al investigated the change of workflow and time in MRI applied to Multiple sclerosis to show that AI is helpful to increase the efficiency of diagnosis. The authors show the details of workflow with and without AI. They also let two radiologists test the workflow on 21 cases. Although the sample size is not very large, this is a meaningful work to present that machine learning software can improve the efficiency of the diagnosis. However, some points need to be improved in the manuscripts.

1. Please spell out the “MRI” for the first occurrence. Some readers may not be familiar with this abbreviation.

2. The way to present time is weird in this manuscript. For example: “4:56 minute” in Line 284 and “82,4 sec” in Line 288 (lines 280 to 293). I suggest the author use seconds only.

3. In Table 1, what is the meaning of the comma? (e.g. is 39,62 means 39.62 or 3962 ?). I suggest the author write in a general way. Otherwise, the reader will be confused.

4. Why the number of machine learning in Table 1 is different from other methods? If the result is generated from the same  21 samples, the results will be more convincing.

5. Please increase the resolution of the figures (Figure 3). I cannot see the text clearly in the figure.

Comments on the Quality of English Language

Please use a more general way to write results (see suggestions above).

Author Response

Dear reviewer 3, Dear ‘healthcare’ team,

we hope this letter finds you well. We are writing to submit the revised version of our manuscript “Changes in MRI Workflow of Multiple Sclerosis after Introduction of an AI-software: a qualitative study", which was previously submitted for consideration to healthcare. We appreciate the opportunity to address the feedback provided by the reviewers. My team and I would like to express our gratitude for the constructive comments and suggestions offered by the reviewers. We have carefully reviewed the remarks and suggestions. As there were overlaps in the points of critique, we have consolidated the addressed issues into a single shared document. We hope this meets with your approval. In the attached PDF file is a modified manuscript version with marked text changes. In response to the feedback, we have meticulously revised the manuscript, incorporating the following changes:

  1. Mdbrain (machine learning-algorithm, development/training); reviewers 1, 2, 4, 5:

We have contacted Mediaire GmbH and requested more detailed information about the development and testing of the ML (machine learning) software mdbrain. The methods section has been supplemented with this information (line 104-135), and a corresponding acknowledgment of this contribution has been inserted into the acknowledgments section. We kindly request to consider that mdbrain is a commercial software, and due to proprietary reasons, deeper insights into its development are currently not available.

  1. Mdbrain options/fine-tuning; reviewer 1:

There were no further configuration options for the user. We added comment for clarity in line 138-139.

  1. Staff training; reviewer 4:

There was no dedicated training of the staff for this study. The staff was however familiar with the analyzing process with and without ML software in the clinical working routine (line 180-182).

  1. Pixel size (line 115-117); reviewer 1:

Pixel sizes can be calculated with the matrix size and FOV provided. If we should provide it, despite its redundancy, please let us know.

  1. Time of ML analysis <5 min; reviewer 1:

We can confirm this time frame for volumetry (specified in manuscript). According to the manufacturer's specifications, the evaluation time varies depending on the technical setup, such as server specifications, especially the built-in graphics processor, and network capacities. The pure export process from the PACS to the local server takes approximately 1:25 minutes. Subsequently, there is a configurable latency period during which the server waits for any subsequent input (currently set to 30 seconds), so that the analysis process starts approximately 2:05 minutes after the initial send command. On the universally available web interface, one obtains the volumetric result after about 4:15 minutes, the lesion report including longitudinal comparison after approximately 7:18 minutes. The mirroring of the report sheets into the PACS is completed after approximately 8:00 minutes.

  1. Table 1 and Figure 3 showing the same information (line 301); reviewer 1:

With the plot of time versus number of lesions, we want to make clear that there is no correlation between the number of lesions and reporting time, neither when using ML, nor without using ML. It also demonstrates, that a high lesion count leads to a longer assessment time. As you point out, we show data that is redundant in the table and box plot. We w consider the box plot to be a very impressive image to emphasize the significant difference in time between diagnosis with and without the use of ML. We believe that it enables the readers to grasp the core message at a glance.

  1. Accuracy of ML vs. ‘gold standard’ or benchmark; reviewers 1, 2, 4:

We share the appreciation for a quality comparison between human-based findings and those generated by ML, and we aim to conduct such a quality assessment for a subsequent publication.

  1. Lack of validation of ML-findings; reviewer 2:

Please take into consideration that this is a qualitative study with a different approach. In our daily work routine, all findings (including those implemented through ML) are analyzed by a neuroradiologist before final approval and corrected if necessary. The objective validation of these results is still pending and will be published at the end of this year.

  1. Statistical tools rationale; reviewer 4:

Since we analyzed paired measurement data, the sample size was larger than 20 and the data in the Q-Q diagram showed a good approximation to a normal distribution, we opted for a parametric test - the paired t-test - to analyze the mean differences between the measurement times with and without the use of ML. In order to assume a deviation from the normal distribution requirement of the paired t-test, we also carried out non-parametric tests for paired samples (Wilcoxon signed-rank test and sign test), that also lead to a significant difference in the median value between the findings with and without ML. We have added a passage on the selection of the paired t-test, and thus hope to have explained the test selection more transparently (line 174-176).

  1. Abbreviations; reviewer 3:

We have added an abbreviation list and included it in the supplementary materials. However, we are open to positioning it elsewhere if preferred.

  1. Time presentation; reviewer 3:

We have now standardized the information in minutes to seconds.

  1. Commas in table 1; reviewer 3:

We apologize for the confusion. In German, the comma marks the decimal places and as our statistic programs work with German input values, outputs are also produced with commas. We did a reformatting into the English conventions and have now adjusted the numerical data in the text and table accordingly.

  1. Different numbers in Table 1 for the ML and non-ML group; reviewer 3:

In the non-ML group 5 radiologists analyzed 5 MRIs each randomized out of a pool of 25 complex MRIs. To prevent selection-bias, we randomized the MRIs for each radiologist. Due to randomization, 4 MRIs were analyzed twice (by different radiologists) and 4 were not chosen at all. The ML group analyzed the MRIs from the non-ML group, so they had only 21 MRIs to analyze. We have now added a passage to the methodology to better clarify the difference in the number of time measurements between MRIs with and without ML (line 164-169).

  1. Resolution of the figure (Figure 3); reviewer 3:

We have seen that the resolution of the images in the .pdf is indeed very poor. We have inserted the images again in higher resolution in the Word document and checked the resolution, when transferring them to a pdf-file. We hope that the images are now actually displayed in a better quality. Please let us know, if there are remaining image quality issues.

  1. Interview selection criteria; reviewer 4:

Paragraph provided (line 92-97).

  1. MDPI format issues; reviewer 5:

The manuscript was created according to the ‘healthcare’-template provided. The categories from the template were utilized.

  1. Display of dataset/data collection; reviewer 5:

The primary data collection consisted of the described interviews (qualitative study). We do not consider a graphical representation to be useful in this context.

  1. Drawbacks or difficulties in using the ML data flow; reviewer 4:

There weren't many difficulties, apart from the initial server setup and the associated administrative challenges. Potential vulnerabilities of the ML software must be considered in the final assessment, as mentioned already in the manuscript. Additional small paragraph provided (line 375-387).

  1. Some figures being put in paper and some in supplementary materials; reviewer 5:

We cannot comprehend the confusion. The main graphics are in the main manuscript, and supplementary graphics, such as the sample reports, are in the supplements.

  1. Enhanced clarity/readability; reviewer 2:

In addition to the aforementioned points, we have attempted to enhance clarity and readability. If there are still unclear passages or ambiguous phrasings in your opinion, we kindly ask you to specifically point them out, and we will be happy to make improvements.

We hope that the revised version aligns with the standards of excellence upheld by healthcare and will be of interest to your readership.

Thank you once again for the opportunity to revise and resubmit our manuscript. I look forward to hearing from you regarding the outcome of the review process. Please do not hesitate to contact me if you require any further information or clarification.

Sincerely,

Eiko Rathmann

Reviewer 4 Report

Comments and Suggestions for Authors

Reviewer Comments:

Major/Minor Comments

1. How were the interview selection criteria chosen? Please properly elaborate on it.

2. How the data set for ML was trained? Please provide the process in a flowchart.

3. How the staff conducting the research using the ML software and algorithms were trained to do the task?

4.What were the drawbacks or difficulties in using the ML data flow, please provide a paragraph.

5. What statistical tools and inferences were used to analyze the data of the research?

6. Also, provide a rationale for using these statistical tests for this research.

7. Did you check the validity of your ML results? Please provide an overview of how you have improved your results according to it.

Author Response

Dear reviewer 4, Dear ‘healthcare’ team,

we hope this letter finds you well. We are writing to submit the revised version of our manuscript “Changes in MRI Workflow of Multiple Sclerosis after Introduction of an AI-software: a qualitative study", which was previously submitted for consideration to healthcare. We appreciate the opportunity to address the feedback provided by the reviewers. My team and I would like to express our gratitude for the constructive comments and suggestions offered by the reviewers. We have carefully reviewed the remarks and suggestions. As there were overlaps in the points of critique, we have consolidated the addressed issues into a single shared document. We hope this meets with your approval. In the attached PDF file is a modified manuscript version with marked text changes. In response to the feedback, we have meticulously revised the manuscript, incorporating the following changes:

  1. Mdbrain (machine learning-algorithm, development/training); reviewers 1, 2, 4, 5:

We have contacted Mediaire GmbH and requested more detailed information about the development and testing of the ML (machine learning) software mdbrain. The methods section has been supplemented with this information (line 104-135), and a corresponding acknowledgment of this contribution has been inserted into the acknowledgments section. We kindly request to consider that mdbrain is a commercial software, and due to proprietary reasons, deeper insights into its development are currently not available.

  1. Mdbrain options/fine-tuning; reviewer 1:

There were no further configuration options for the user. We added comment for clarity in line 138-139.

  1. Staff training; reviewer 4:

There was no dedicated training of the staff for this study. The staff was however familiar with the analyzing process with and without ML software in the clinical working routine (line 180-182).

  1. Pixel size (line 115-117); reviewer 1:

Pixel sizes can be calculated with the matrix size and FOV provided. If we should provide it, despite its redundancy, please let us know.

  1. Time of ML analysis <5 min; reviewer 1:

We can confirm this time frame for volumetry (specified in manuscript). According to the manufacturer's specifications, the evaluation time varies depending on the technical setup, such as server specifications, especially the built-in graphics processor, and network capacities. The pure export process from the PACS to the local server takes approximately 1:25 minutes. Subsequently, there is a configurable latency period during which the server waits for any subsequent input (currently set to 30 seconds), so that the analysis process starts approximately 2:05 minutes after the initial send command. On the universally available web interface, one obtains the volumetric result after about 4:15 minutes, the lesion report including longitudinal comparison after approximately 7:18 minutes. The mirroring of the report sheets into the PACS is completed after approximately 8:00 minutes.

  1. Table 1 and Figure 3 showing the same information (line 301); reviewer 1:

With the plot of time versus number of lesions, we want to make clear that there is no correlation between the number of lesions and reporting time, neither when using ML, nor without using ML. It also demonstrates, that a high lesion count leads to a longer assessment time. As you point out, we show data that is redundant in the table and box plot. We w consider the box plot to be a very impressive image to emphasize the significant difference in time between diagnosis with and without the use of ML. We believe that it enables the readers to grasp the core message at a glance.

  1. Accuracy of ML vs. ‘gold standard’ or benchmark; reviewers 1, 2, 4:

We share the appreciation for a quality comparison between human-based findings and those generated by ML, and we aim to conduct such a quality assessment for a subsequent publication.

  1. Lack of validation of ML-findings; reviewer 2:

Please take into consideration that this is a qualitative study with a different approach. In our daily work routine, all findings (including those implemented through ML) are analyzed by a neuroradiologist before final approval and corrected if necessary. The objective validation of these results is still pending and will be published at the end of this year.

  1. Statistical tools rationale; reviewer 4:

Since we analyzed paired measurement data, the sample size was larger than 20 and the data in the Q-Q diagram showed a good approximation to a normal distribution, we opted for a parametric test - the paired t-test - to analyze the mean differences between the measurement times with and without the use of ML. In order to assume a deviation from the normal distribution requirement of the paired t-test, we also carried out non-parametric tests for paired samples (Wilcoxon signed-rank test and sign test), that also lead to a significant difference in the median value between the findings with and without ML. We have added a passage on the selection of the paired t-test, and thus hope to have explained the test selection more transparently (line 174-176).

  1. Abbreviations; reviewer 3:

We have added an abbreviation list and included it in the supplementary materials. However, we are open to positioning it elsewhere if preferred.

  1. Time presentation; reviewer 3:

We have now standardized the information in minutes to seconds.

  1. Commas in table 1; reviewer 3:

We apologize for the confusion. In German, the comma marks the decimal places and as our statistic programs work with German input values, outputs are also produced with commas. We did a reformatting into the English conventions and have now adjusted the numerical data in the text and table accordingly.

  1. Different numbers in Table 1 for the ML and non-ML group; reviewer 3:

In the non-ML group 5 radiologists analyzed 5 MRIs each randomized out of a pool of 25 complex MRIs. To prevent selection-bias, we randomized the MRIs for each radiologist. Due to randomization, 4 MRIs were analyzed twice (by different radiologists) and 4 were not chosen at all. The ML group analyzed the MRIs from the non-ML group, so they had only 21 MRIs to analyze. We have now added a passage to the methodology to better clarify the difference in the number of time measurements between MRIs with and without ML (line 164-169).

  1. Resolution of the figure (Figure 3); reviewer 3:

We have seen that the resolution of the images in the .pdf is indeed very poor. We have inserted the images again in higher resolution in the Word document and checked the resolution, when transferring them to a pdf-file. We hope that the images are now actually displayed in a better quality. Please let us know, if there are remaining image quality issues.

  1. Interview selection criteria; reviewer 4:

Paragraph provided (line 92-97).

  1. MDPI format issues; reviewer 5:

The manuscript was created according to the ‘healthcare’-template provided. The categories from the template were utilized.

  1. Display of dataset/data collection; reviewer 5:

The primary data collection consisted of the described interviews (qualitative study). We do not consider a graphical representation to be useful in this context.

  1. Drawbacks or difficulties in using the ML data flow; reviewer 4:

There weren't many difficulties, apart from the initial server setup and the associated administrative challenges. Potential vulnerabilities of the ML software must be considered in the final assessment, as mentioned already in the manuscript. Additional small paragraph provided (line 375-387).

  1. Some figures being put in paper and some in supplementary materials; reviewer 5:

We cannot comprehend the confusion. The main graphics are in the main manuscript, and supplementary graphics, such as the sample reports, are in the supplements.

  1. Enhanced clarity/readability; reviewer 2:

In addition to the aforementioned points, we have attempted to enhance clarity and readability. If there are still unclear passages or ambiguous phrasings in your opinion, we kindly ask you to specifically point them out, and we will be happy to make improvements.

We hope that the revised version aligns with the standards of excellence upheld by healthcare and will be of interest to your readership.

Thank you once again for the opportunity to revise and resubmit our manuscript. I look forward to hearing from you regarding the outcome of the review process. Please do not hesitate to contact me if you require any further information or clarification.

Sincerely,

Eiko Rathmann

Reviewer 5 Report

Comments and Suggestions for Authors

carefully addressedd all comments

Author Response

Dear reviewer 5, Dear ‘healthcare’ team,

we hope this letter finds you well. We are writing to submit the revised version of our manuscript “Changes in MRI Workflow of Multiple Sclerosis after Introduction of an AI-software: a qualitative study", which was previously submitted for consideration to healthcare. We appreciate the opportunity to address the feedback provided by the reviewers. My team and I would like to express our gratitude for the constructive comments and suggestions offered by the reviewers. We have carefully reviewed the remarks and suggestions. As there were overlaps in the points of critique, we have consolidated the addressed issues into a single shared document. We hope this meets with your approval. In the attached PDF file is a modified manuscript version with marked text changes. In response to the feedback, we have meticulously revised the manuscript, incorporating the following changes:

  1. Mdbrain (machine learning-algorithm, development/training); reviewers 1, 2, 4, 5:

We have contacted Mediaire GmbH and requested more detailed information about the development and testing of the ML (machine learning) software mdbrain. The methods section has been supplemented with this information (line 104-135), and a corresponding acknowledgment of this contribution has been inserted into the acknowledgments section. We kindly request to consider that mdbrain is a commercial software, and due to proprietary reasons, deeper insights into its development are currently not available.

  1. Mdbrain options/fine-tuning; reviewer 1:

There were no further configuration options for the user. We added comment for clarity in line 138-139.

  1. Staff training; reviewer 4:

There was no dedicated training of the staff for this study. The staff was however familiar with the analyzing process with and without ML software in the clinical working routine (line 180-182).

  1. Pixel size (line 115-117); reviewer 1:

Pixel sizes can be calculated with the matrix size and FOV provided. If we should provide it, despite its redundancy, please let us know.

  1. Time of ML analysis <5 min; reviewer 1:

We can confirm this time frame for volumetry (specified in manuscript). According to the manufacturer's specifications, the evaluation time varies depending on the technical setup, such as server specifications, especially the built-in graphics processor, and network capacities. The pure export process from the PACS to the local server takes approximately 1:25 minutes. Subsequently, there is a configurable latency period during which the server waits for any subsequent input (currently set to 30 seconds), so that the analysis process starts approximately 2:05 minutes after the initial send command. On the universally available web interface, one obtains the volumetric result after about 4:15 minutes, the lesion report including longitudinal comparison after approximately 7:18 minutes. The mirroring of the report sheets into the PACS is completed after approximately 8:00 minutes.

  1. Table 1 and Figure 3 showing the same information (line 301); reviewer 1:

With the plot of time versus number of lesions, we want to make clear that there is no correlation between the number of lesions and reporting time, neither when using ML, nor without using ML. It also demonstrates, that a high lesion count leads to a longer assessment time. As you point out, we show data that is redundant in the table and box plot. We w consider the box plot to be a very impressive image to emphasize the significant difference in time between diagnosis with and without the use of ML. We believe that it enables the readers to grasp the core message at a glance.

  1. Accuracy of ML vs. ‘gold standard’ or benchmark; reviewers 1, 2, 4:

We share the appreciation for a quality comparison between human-based findings and those generated by ML, and we aim to conduct such a quality assessment for a subsequent publication.

  1. Lack of validation of ML-findings; reviewer 2:

Please take into consideration that this is a qualitative study with a different approach. In our daily work routine, all findings (including those implemented through ML) are analyzed by a neuroradiologist before final approval and corrected if necessary. The objective validation of these results is still pending and will be published at the end of this year.

  1. Statistical tools rationale; reviewer 4:

Since we analyzed paired measurement data, the sample size was larger than 20 and the data in the Q-Q diagram showed a good approximation to a normal distribution, we opted for a parametric test - the paired t-test - to analyze the mean differences between the measurement times with and without the use of ML. In order to assume a deviation from the normal distribution requirement of the paired t-test, we also carried out non-parametric tests for paired samples (Wilcoxon signed-rank test and sign test), that also lead to a significant difference in the median value between the findings with and without ML. We have added a passage on the selection of the paired t-test, and thus hope to have explained the test selection more transparently (line 174-176).

  1. Abbreviations; reviewer 3:

We have added an abbreviation list and included it in the supplementary materials. However, we are open to positioning it elsewhere if preferred.

  1. Time presentation; reviewer 3:

We have now standardized the information in minutes to seconds.

  1. Commas in table 1; reviewer 3:

We apologize for the confusion. In German, the comma marks the decimal places and as our statistic programs work with German input values, outputs are also produced with commas. We did a reformatting into the English conventions and have now adjusted the numerical data in the text and table accordingly.

  1. Different numbers in Table 1 for the ML and non-ML group; reviewer 3:

In the non-ML group 5 radiologists analyzed 5 MRIs each randomized out of a pool of 25 complex MRIs. To prevent selection-bias, we randomized the MRIs for each radiologist. Due to randomization, 4 MRIs were analyzed twice (by different radiologists) and 4 were not chosen at all. The ML group analyzed the MRIs from the non-ML group, so they had only 21 MRIs to analyze. We have now added a passage to the methodology to better clarify the difference in the number of time measurements between MRIs with and without ML (line 164-169).

  1. Resolution of the figure (Figure 3); reviewer 3:

We have seen that the resolution of the images in the .pdf is indeed very poor. We have inserted the images again in higher resolution in the Word document and checked the resolution, when transferring them to a pdf-file. We hope that the images are now actually displayed in a better quality. Please let us know, if there are remaining image quality issues.

  1. Interview selection criteria; reviewer 4:

Paragraph provided (line 92-97).

  1. MDPI format issues; reviewer 5:

The manuscript was created according to the ‘healthcare’-template provided. The categories from the template were utilized.

  1. Display of dataset/data collection; reviewer 5:

The primary data collection consisted of the described interviews (qualitative study). We do not consider a graphical representation to be useful in this context.

  1. Drawbacks or difficulties in using the ML data flow; reviewer 4:

There weren't many difficulties, apart from the initial server setup and the associated administrative challenges. Potential vulnerabilities of the ML software must be considered in the final assessment, as mentioned already in the manuscript. Additional small paragraph provided (line 375-387).

  1. Some figures being put in paper and some in supplementary materials; reviewer 5:

We cannot comprehend the confusion. The main graphics are in the main manuscript, and supplementary graphics, such as the sample reports, are in the supplements.

  1. Enhanced clarity/readability; reviewer 2:

In addition to the aforementioned points, we have attempted to enhance clarity and readability. If there are still unclear passages or ambiguous phrasings in your opinion, we kindly ask you to specifically point them out, and we will be happy to make improvements.

We hope that the revised version aligns with the standards of excellence upheld by healthcare and will be of interest to your readership.

Thank you once again for the opportunity to revise and resubmit our manuscript. I look forward to hearing from you regarding the outcome of the review process. Please do not hesitate to contact me if you require any further information or clarification.

Sincerely,

Eiko Rathmann

Round 2

Reviewer 2 Report

Comments and Suggestions for Authors

The study fails to provide adequate details regarding the integration and evaluation of the ML software (mdbrain®) into the MRI workflow. Without clear steps and procedure of the software’s implementation process, including criteria for evaluation and clear explanation about the underlying algorithms, it is impossible to assess the validity and reliability of the presented findings in the study. This lack of details raises concerns about the ethical consideration of the study and undermines the scientific sound.

The study findings are only supported by a limited and narrow sample size, it limits the external validity of the results and raising concerns about the generability of the model. Without a more diverse sample representation with different data and patient populations, we can’t conclude the applicability of the model beyond its specific context.

In summary, the paper is unfit for the publication due to limited external validity, inadequate methodology explanation, incomplete assessment of limitations and insufficient supplementary materials. Additionally, the paper suffers from problems in presentation and structure. It contains sections that are difficult to follow due to a disorganized flow of content. To ensure the scientific rigor and credibility of the research, these serious flaws must be fixed.

Comments on the Quality of English Language

Overall, while the quality of English in the paper is satisfactory, there is area for improvement in terms of enhancing readability, refining sentence structures for clarity, and need to ensure the consistency in technical terminology usage.

Author Response

Dear Editors and Reviewers,

thank you again for taking the time to further improve our manuscript!

 Editor 3's note regarding the conversion of commas to periods should have been included in the last revision already. I apologize for that oversight. The corrected table is now included to ensure that the information provided does not unnecessarily confuse the reader.

Unfortunately, there are still uncertainties regarding the study design that I would like to address. Our work is a qualitative study, as indicated in the title and the last paragraph of the introduction. To further emphasize that in the manuscript, I have added a reference in the last paragraph (https://www.ncbi.nlm.nih.gov/books/NBK470395/) that explains the general concept of qualitative studies. We understand that qualitative studies are rare in medicine. In social sciences, psychology, and business economics qualitative studies are a common tool aiming to extract hypotheses for further quantitative studies. For our scenario of implementing a new technology into everyday work, we consider the interview-based approach to be a valuable contribution to the current topic of AI-assisted diagnosis in radiological practice.

The extensive interviews and the information derived from them constitute the actual core of our work. From these, descriptions of the different workflows and the crucial differences with potential for workflow optimizations or process errors were elaborated, and ultimately, key statements from the employees regarding the process change were obtained. The time measurements introduced through a small sample size are an addition to the core of the work. The goal of our study was not to develop or validate a model. Therefore, it is not possible to provide a confusion matrix.

In our last revision, we provided additional information about the underlying algorithm: “Mdbrain® uses a deep convolutional neural network (U-Net) trained with annotated ground truth data of more than 1,000 heterogenous patient data sets from different scanner types and sequences. The volumes of these regions are then quantified, and the corresponding percentiles are calculated by comparing the measured patient’s volumes to the volumes of a healthy population (8,500 healthy people) with the same covariates (age, sex, and total intracranial volume) “. We apologize that it is not possible to provide more precise information about the underlying algorithm, as it is a proprietary product approved according to the European Medical Device Directive and DIN EN ISO standards.

Linguistic enhancements have been made to improve readability and comprehensibility.

We would like to apologize for any misunderstandings and thank you once again for the revision suggestions for our manuscript.

Kind regards,

Eiko Rathmann and the author team

Reviewer 3 Report

Comments and Suggestions for Authors

The issues have been improved. 

Comments on the Quality of English Language

The issues have been improved. The author still uses commas in the manuscript. But I suggest using dots rather than commas in Table 1 and line 353. Most readers may be confused about this.

Author Response

(The authors gave the same response as above.)
